# The Influence of Chemical Component Distribution on the Radiometric Properties of Particle Aggregates

**Yizhan Chai, Zhen Yang * and Yuanyuan Duan**

Key Laboratory of Thermal Science and Power Engineering of Education Ministration, Beijing Key Laboratory for CO2 Utilization and Reduction Technology, Tsinghua University, Beijing 100084, China; chaiyz14@mails.tsinghua.edu.cn (Y.C.); yyduan@mail.tsinghua.edu.cn (Y.D.)

* Correspondence: zhenyang@tsinghua.edu.cn

**Abstract:** The radiometric properties, including the extinction efficiency, absorption efficiency, scattering efficiency, and asymmetric parameter values of particle aggregates consisting of multiple chemical components are critical in industry and nature. This article aims to analyze the influence of chemical component distribution on these radiometric properties. The particle aggregates are generated by a diffusion-limited aggregate method by which spherical particles are stuck together randomly. The particle aggregates have two components with a major component of a fixed refractive index and a minor component of a changed refractive index. The radiometric properties are calculated using a multi-sphere T-matrix (MSTM) method for particle aggregates with different particle radii and with refractive indices, distributions of components, and volume fractions of the minor component. The results show that the chemical component distribution influences the radiometric properties of the particle aggregate. Evenly spreading the strong absorptive minor component into each particle, compared to concentrating it in a few particles, can raise the absorption efficiency, reduce the scattering efficiency, and ultimately reduce the extinction efficiency of the aggregate. For aggregates with major and minor components in different particles, a similar effect is shown when spreading the minor component particles evenly in the aggregate relative to gathering them in one part of the aggregate.

**Keywords:** radiometric properties; particle aggregate; MSTM; chemical component

## 1. Introduction

The radiometric properties of particle aggregates mainly include extinction efficiency, absorption efficiency, scattering efficiency, and asymmetric parameter [1–4]. It is crucial in numbers of science and engineering disciplines, including coal combustion [5,6], the concrete aggregate industry [7], atmospheric and environmental sciences, materials engineering [8], plasmatic photo-thermal therapy [9], and nuclear reactor safety [10]. For example, cosmic dust is a typical type of particle aggregate; its radiometric properties are related to the formation of the universe [11]. During plasmatic photo-thermal therapy, AuNP particle aggregates interact with near-infrared (NIR) radiation. Absorption efficiency and scattering efficiency need to be determined accurately so that the therapy uses a proper power rate in case of human tissue damage [9].

The radiometric properties of particle aggregates are fundamentally governed by the macroscopic Maxwell's wave equations (MWEs). The microscopic-level complexity of particle aggregates makes a direct solution for MWEs unfeasible. Effective-medium approximations (EMAs) have been widely used to model the radiometric properties of homogeneous substances [12–22]. The unparalleled simplicity of EMAs leads to a simplified calculation. Even for complex heterogeneous substances, EMAs are still used by treating substances as homogeneous and having a refractive index computed

with a phenomenological mixing rule such as the Lorentz–Lorenz, Bruggeman, and Maxwell–Garnett formulas [23,24]. Mackowski and Mishchenko [15] studied the radiometric properties of a slab of compact, pure, spherical particles with EMAs and found that the results agreed closely with measurements. Mishchenko [22] compared a mixture of two materials with generic refractive indices of 1.33 and 1.55 and indicated that conventional EMAs could be improved by freely varying the real and imaginary parts of the refractive index of the mixture, at least when the inclusions had size parameters lower than 0.5. In most cases, using EMAs can lead to acceptable results [15,25,26]. However, since an EMA ignores the structure and chemical component distribution inside particle aggregates, it may not be able to predict radiometric properties correctly or accurately for heterogeneous particle aggregates with a complex structure and a chemical component distribution. Bohren [27] mentioned "What is at issue here is not which effective-medium theory is applicable to a given heterogeneous medium, but whether or not any of them are."

For spherical particle aggregates with complex structures, the multi-sphere T-matrix (MSTM) method is used to calculate radiometric properties [15,22,28,29]. The MSTM performs a direct simulation of EM wave propagation in large-scale systems of spheres and considers particle position, size, and chemical components [30,31]. With the MSTM method, researchers can validate the accuracy of EMAs. Kolokolova [3] used the MSTM to study the effect of the hierarchical structure of particle aggregates on radiometric properties of cometary dust particles; the results showed the importance of the aggregate structure to radiometric properties, and this could not be revealed by EMAs. Fan [32] studied the effect of structure morphology on optical properties of soot aggregates using MSTM; similar to Kolokolova's results, the aggregate structure had an important influence on the radiometric properties.

The chemical component in a particle aggregate is essential to radiometric properties. Kahnert [19] indicated that the chemical component inhomogeneity was potentially important and needed to receive more attention when the radiometric properties of mineral particles of hematite inclusions are calculated with the MSTM. Realistic morphological black carbon fractal aggregates coated with chemically heterogeneous sulfate were also investigated by Kahnert [16,33]. In such a complex chemical component system, EMA calculations deviates significantly from MSTM calculations. Kolokolova [3] used the MSTM to study the effect of real structure material on radiometric properties of particle aggregates to support the study of cometary dust particles; however, the chemical component of the heterogeneous material is considered uniformly distributed. Up to now, how this chemical component influences the radiometric properties of particle aggregates has remained unclear. In fact, this component's distribution in particle aggregates in real matter is complicated [34]. One SEM image of ash deposits from biomass-fired boilers clearly shows that this component is not evenly distributed. Hematite and other minerals have been found to be distributed as small clusters in ash deposits [35]; this chemical component can aggregate into small clusters at the same melting point.

The purpose of this work is to study the influence of chemical component distribution on the radiometric properties of particle aggregates. The chemical component distribution is important to the radiometric properties, and its influence needs to be properly understood. Particle aggregates are modeled by randomly-placed spherical particles with the distributions of two chemical components, and the radiometric properties of the aggregates are calculated using the MSTM. The particle size, refractive indices, and volume fractions of the minor chemical component are inspected. The influence of the chemical component distribution in particle aggregates on radiometric properties is analyzed.

## 2. Methods

### 2.1. Particle Aggregate

In this work, particle aggregates are composed of single-sized spherical particles, as shown in Figure 1. The reason for choosing spherical particles is that large numbers of real particle aggregates can be regarded as composed of spherical particles, and the radiometric properties of spherical particles

are relatively easy to calculate, so this work can be focused on the influence of chemical component distribution in particle aggregates.

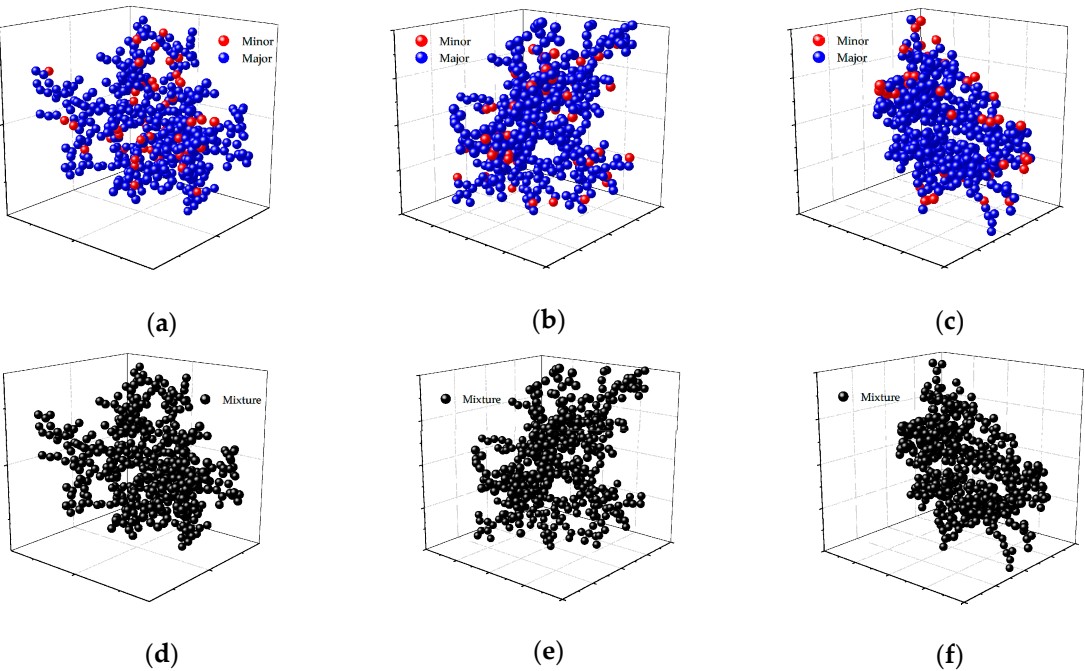

**Figure 1.** Particle aggregates composed by spatially randomly distributed spherical particles of uniform size. Mode A: chemical component concentrated in a few particles; Mode B: chemical components evenly distributed in each particle. (**a**) and (**d**) are of the same structure, as are (**b**) and (**e**) as well as (**c**) and (**f**).

The particle aggregates were generated by a diffusion-limited aggregate (DLA) [36,37] method by which particles are initially randomly distributed in space and then randomly walk until they become stuck to each other. The particle radius $R$ was set to 2–5 μm; this particle radius range is common for a number of real particles, such as comet dust [11,38,39] or coal smoke particles [40,41]. The particle number in the aggregates was set to 500. Increasing the particle number was found not to significantly change the radiometric properties of the aggregate, while it exponentially increases the computation time; moreover, the influence of chemical component distribution can be sufficiently revealed with the aggregates of 500 particles. The particle aggregates have two chemical components, namely the major component and the minor component. For the detailed DLA process, the seed particle was placed at the origin point. After that, the initial position of the particle was randomly generated on a spherical surface with a radius 100 times that of the particle. The random walking process was applied to the particle until it hit the seed particle; the walking step length was set to 0.1 times the radius of the particle. After hitting, a minor adjustment was made based on the two particles' radii to make sure the two particles contacted each other. This process was repeated until an aggregate structure was formed. Using similar processes, different aggregate structures were generated.

To investigate the influence of chemical component distribution, two component distribution modes were introduced as Mode A and Mode B. In Mode A, a particle is composed either of a major component or a minor component; the minor component is concentrated in a few particles, as it is in many natural agglomerations and particles [42,43]. In Mode B, all particles are the same, and the major and minor components are evenly mixed in each particle. Figure 1a,d show a particle aggregate in Mode A and Mode B; the two modes have different distributions of the minor component. Figure 1b,e show a second aggregate of Mode A and Mode B; Figure 1c,f show a third.

The volume fraction of the minor component, *f*, is calculated by dividing the volume of the minor component by the volume of both components in the aggregate. In this study, *f* = 1 – 25%, a relatively wide range of the volume fractions of the minor component.

In Mode A, the particles of the minor component may have various positions in the aggregate, which may affect the radiometric properties of the aggregate. When the minor component particles are randomly positioned, it is said that the minor component particles have a random position pattern. For a particle aggregate, three random position patterns of the minor component particles are considered, as shown in Figure 2d–f; the radiometric properties of the aggregate with the three patterns are calculated, and the average radiometric properties are used as the radiometric properties of the particle aggregate with the random position pattern, thus minimizing the impact of certain patterns on the calculation.

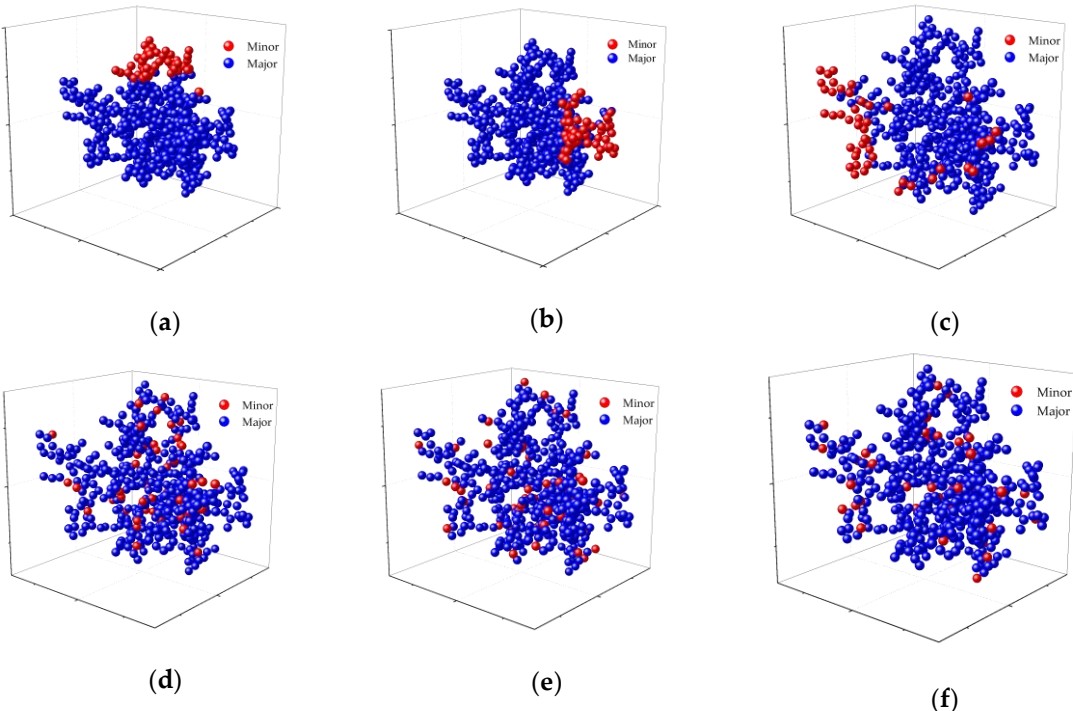

**Figure 2.** Different position patterns of the minor component particles: particles gathered in minor spheres (red) in the upper (**a**), in the right (**b**), and in the front(**c**) and randomly positioned particles in (**d**–**f**).

Aside from the random pattern, other position patterns of the minor component particles are also considered for comparison: the pattern of minor component particles in the upper, right, or front part of the aggregate, as respectively shown in Figure 2a–c.

*2.2. Refractive Index*

The MSTM uses the refractive index of each particle to calculate the radiometric properties of the particle aggregate. For Mode A and Mode B particle aggregates, the refractive indices are determined in different ways.

2.2.1. Mode A

In Mode A, each particle has a pure chemical component. The optical refractive index of major component particles $m_1$ is 1.6, the typical value of calcite or dolomite [44]. The optical refractive indices of the minor component particles $m_2$ are set to three different values, i.e., $m_2$ = 3.102 + 0.0925i [45], 1.46 + 2.75i [45], and 0.15 + 3.15i [23], respectively, for cases where the real part is larger than the imaginary part, the real part and the imaginary part is large, and the real part is smaller than the

imaginary part. These three values respectively correspond to that for hematite, quartz at certain IR wavelengths, and gold in the visible wavelengths [46]. The real part of the refractive index indicates the speed of the light travelling in the medium, and the imaginary part indicates the absorption ability of the medium.

### 2.2.2. Mode B

In Mode B, the particles in the aggregate are from even mixtures of the two components. For such aggregates, the EMA method is suitable for calculating the refractive index of the spherical particles in the aggregates. The Maxwell–Garnett is the most frequently used EMA for spherical particles composed of a minor component and a major component. The effective refractive index $m_{\text{eff}}$ of the spherical particles is [19,47]:

$$m_{\text{eff}} = \sqrt{m_1 \frac{m_1^2 \cdot 2(1-f) + m_2^2(1+2f)}{m_1^2(2+f) + m_2^2(1-f)}} \tag{1}$$

where refractive index and volume fraction mixed in a major chemical component of $m_1$ and $m_2$ are the refractive indices of the major and minor components, respectively; $f$ is the volume fraction of the minor component.

The refractive indices $m_1$ and $m_2$ are the same as those in Mode A. The real and imaginary parts of the effective refractive index $m_{\text{eff}}$ for different $m_2$ values are shown in Figure 3.

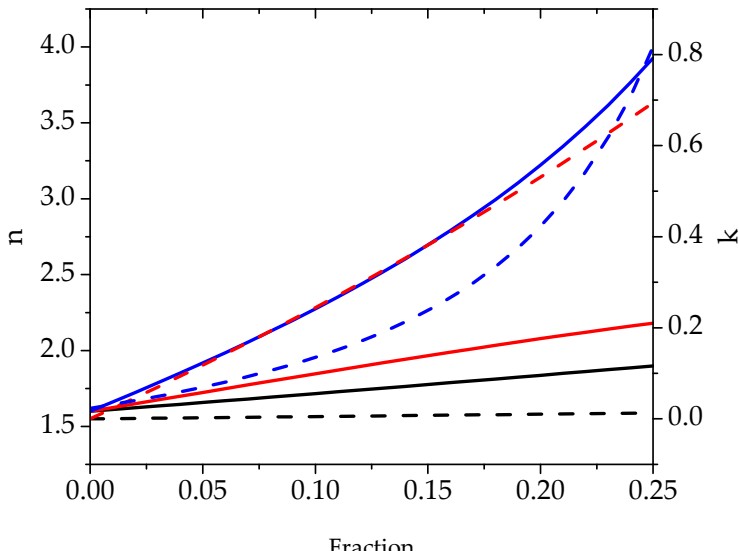

**Figure 3.** Effective refractive indices computed with the Maxwell–Garnett (MG) method (solid lines: the real part of refractive index n; dashed lines: the imaginary part of refractive index k. Black line: $m_2$ = 3.102 + 0.0925i; red line: $m_2$ = 1.46 + 2.75i; blue line: $m_2$ = 0.15 + 3.15i).

The real part of the effective refractive indices, *n*, increases with volume fraction for all values of $m_2$. The imaginary part of effective refractive indices, *k*, also increases with volume fraction for all values of $m_2$.

### 2.3. Radiometric Properties Computation

The MSTM is used to calculate the radiometric properties of the particle aggregates in this study. The MSTM is one of the most efficient and accurate models used to numerically calculate the random-orientation absorption properties of spherical particle aggregates with non-overlapping surfaces. It employs the addition theorem of vector spherical wave functions to account for mutual

interactions among the multiple-sphere system, and its T-matrix for deriving particle absorption properties can be achieved from the individual spheres [30]. The robust and popular MSTM code has already been widely utilized for many numerical investigations of the absorption properties of fractal aggregates [3,28,48]. The MSTM assumes that the T-matrices of all particles in the aggregate are the superposition of individual contributions from each particle. It can be expressed as:

$$\mathbf{E}_{sca}(\mathbf{r}) = \sum_{j=1}^{N} E_{sca,j}(r) \tag{2}$$

where $\mathbf{E}_{sca,j}$ is the scattered flied for sphere $j$. The $\mathbf{r}$ is the vector connected to the observation point and base reference system. The scattering particles interact with each other and are related to the scattering field. Thus, the excitation field excited on each particle can be described as the superposition of the external incident field and the associated field scattered by other particles:

$$\mathbf{E}_{\text{inc},j}(\mathbf{r}) = \mathbf{E}_{\text{inc},0}(\mathbf{r}) \sum_{l=1,l\neq j}^{N} E_{sca,l}(r), j = 1, \dots, N \tag{3}$$

where $\mathbf{E}_{\text{inc},j}$ is the incident flied for sphere $j$, and $\mathbf{E}_{\text{inc},0}$ is the external incident flied. With the translation of particle $i$ to particle $j$, the T-matrix for the particle aggregates is calculated. The aggregate T-matrix can be used to compute the amplitude matrix for a fixed orientation of aggregates and can be input into analytical procedures for averaging scattering characteristics over orientations [30,49].

In this article, the MSTM is used to calculate the radiometric properties of particle aggregates. A FORTRAN-90 Multiple Sphere T-matrix (MSTM) code developed by Mackowski and Mishchenko [50] is utilized. The latest version of the MSTM code was released in 2013 (available at [51]). The computations were taken on a Dell workstation with 2 Intel Xeon Gold 5118 CPUs (2.30GHz), $2 \times 128$ GB memory. Each computation took about 4 h to finish.

## 3. Results and Discussion

### 3.1. Distribution Mode

Figure 4 shows the spectrum-averaged radiometric properties, including the extinction efficiency, absorption efficiency, scattering efficiency, and asymmetric parameter, varied with particle radius for particle aggregates of Mode A and Mode B with different minor components. In Mode A, the minor component particles are randomly positioned. The spectrum-averaged radiometric properties are averaged at wavelengths of 6.5–10 μm.

As seen in the figure, the curves for Mode A are different from those for Mode B, showing that the distribution of components does affect the radiometric properties to various extents.

As for the absorption efficiency (Figure 4b), Mode A has lower values than Mode B when $m_2 = 0.15 + 3.15i$ and $m_2 = 1.46 + 2.75i$. This shows that, when the minor component has a larger imaginary part of the refractive index or is more absorptive compared to the major component ($m_1 = 1.6$), concentrating the minor component in a few particles makes the particle aggregate less absorptive, and spreading the minor component into each particle makes the particle aggregate more absorptive. In addition, the absorption efficiency with $m_2 = 1.46 + 2.75i$ is greater than that with $m_2 = 0.15 + 3.15i$ for Mode A and Mode B. The absorption efficiency of Mode A is 60.3% lower than that of Mode B when $m_2 = 1.46 + 2.75i$ and is 90.3% lower when $m_2 = 0.15 + 3.15i$. It should be noted that the absorption efficiency of a particle aggregate is mainly influenced by the imaginary part of the refractive index and that, when $m_2 = 1.46 + 2.75i$ and $m_2 = 0.15 + 3.15i$, the imaginary part of the refractive index is significantly larger than it is when major component $m_1 = 1.6$ (no adsorption). For Mode A, only the minor component particles contribute to absorption; however, for Mode B, all particles in the aggregate are absorptive, since the mixed material has a non-zero imaginary part of the refractive index, resulting in greater absorption efficiency than Mode A. Moreover, absorption

grows exponentially with optical length [1,52], and the particles in Mode B have a larger real part of the refractive index compared to the major component, and this increases the effective optical length in each particle and, therefore, makes the aggregate more absorptive. When $m_2 = 3.102 + 0.0925i$, the imaginary part nears that of the major component (which is zero), so the minor component is close to the major component in terms of absorption; under such a condition, the distribution of components becomes much less important to the absorption efficiency of the aggregate since both components are close in terms of absorption, so Mode A and Mode B become similar in terms of absorption efficiency, as evident by the relatively small difference of 8% between Mode A and Mode B. However, as the particle radius increases, the absorption efficiency between Mode A and Mode B changes: it decreases in Mode A and increases in Mode B. This may be attributed to the different distributions of the components between Mode A and Mode B.

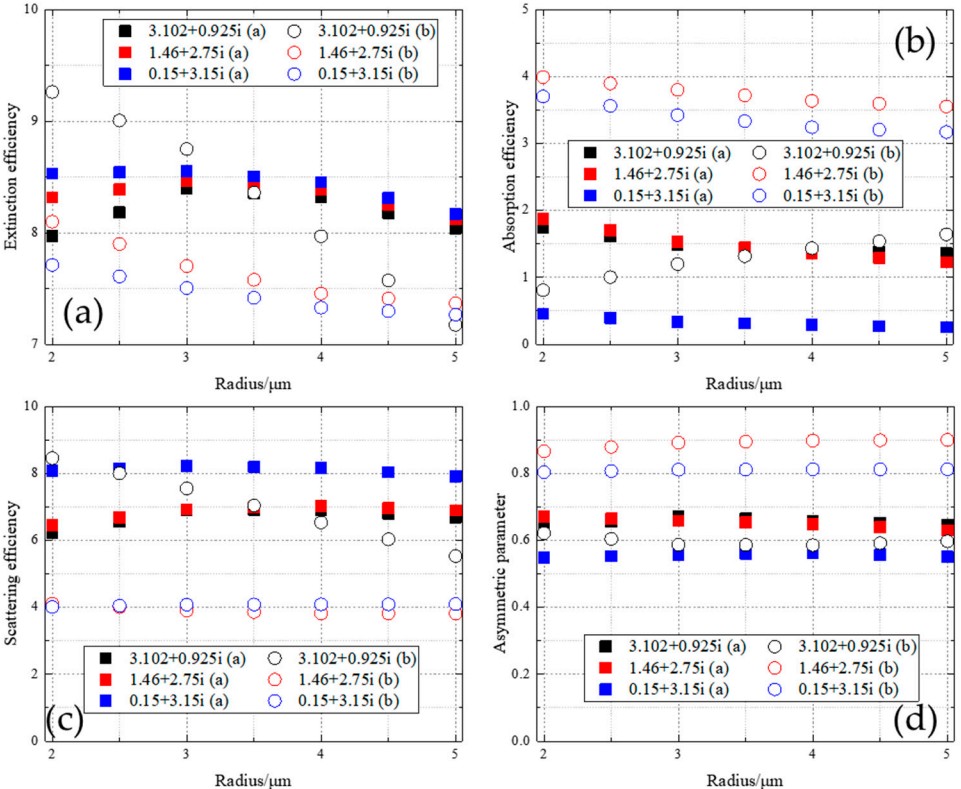

**Figure 4.** Radiometric properties at various particle radii for particle aggregates of different minor components at $f = 25\%$ (square symbol: Mode A, with the minor component particles randomly positioned; circle symbol: Mode B; black symbol: minor component $m_2 = 3.102 + 0.0925i$; red symbol: minor component $m_2 = 1.46 + 2.75i$; blue symbol: minor component $m_2 = 0.15 + 3.15i$). (**a**) Extinction efficiency; (**b**) absorption efficiency; (**c**) scattering efficiency; (**d**) asymmetric parameter.

For the scattering efficiency (Figure 4c), Mode A has greater values compared to Mode B when $m_2 = 0.15 + 3.15i$ or $m_2 = 1.46 + 2.75i$. This means, when the minor component has a large imaginary part of the refractive index, concentrating the minor component in a few particles causes more scattering in the aggregate compared to when it is evenly spread into each particle, indicating the important role that a few scattering particles play on the scattering efficiency of the aggregate. Scattering mostly occurs on the interface of the two chemical components [53]. In Mode B, the EMA method treated the two chemical components as evenly mixed and therefore artificially decreased the interface of the two chemical components such that scattering was reduced. When $m_2 = 3.102 + 0.0925i$, Mode A and Mode B are similar in scattering efficiency, probably because both components have similar refractive index imaginary parts, so the distribution of components is less important.

Figure 4d shows the asymmetric parameter. A smaller asymmetry parameter indicates less forward scattering compared to backward scattering by the particle aggregate. When $m_2 = 0.15 + 3.15i$ or $m_2 = 1.46 + 2.75i$, Mode A has lower asymmetric parameter values compared to Mode B, showing that the former mode has a stronger backward scattering. It should be remembered that Mode B increases the absorption of the entire particle aggregate. For forward scattering, the effective absorption optical length is less than that of backward scattering, so the increase in the absorption of the aggregate cause a greater reduction in backward scattering. This results in an increase in the relative proportion of forward scattering to total scattering, namely the asymmetry parameter. When $m_2 = 3.102 + 0.0925i$ and the imaginary parts of the refractive indices of the minor component are close to those of the major component (which is zero), Mode A and Mode B have similar asymmetric parameter values, showing that the distribution mode of the component is much less important when the minor component is close in absorption to the major component.

Figure 4a shows the extinction efficiency, which is the sum of the absorption efficiency and the scattering efficiency. As seen in Figure 4a–d, the radiometric properties of all particle aggregates are generally insensitive to the change in particle radius, except the absorption efficiency, scattering efficiency, and extinction efficiency of Mode B with $m_2 = 3.102 + 0.0925i$. When $m_2 = 3.102 + 0.0925i$, the minor component has a relatively long optical length due to the large real part of its refractive index. In Mode B, the minor component is evenly scattered into each particle of the aggregate, which increases the optical length of the aggregate; as the particle radius increases, the optical length of the aggregate increases relatively significantly, causing more absorption as the incoming radiation passes through the aggregate and hence the absorption efficiency increases with the particle radius. The increase in the absorption of radiation also causes less radiation to be scattered and therefore reduces the scattering efficiency of the aggregate as the particle radius increases, as evidenced by the line for Mode B with $m_2 = 3.102 + 0.0925i$ in Figure 4c.

*3.2. Minor Particle Position*

In the particle aggregate of Mode A, the positions of the minor component particles also influence the radiometric properties. Figure 5 provides spectrum-averaged radiometric properties of the particle aggregate with different position patterns of the minor component particles. The spectrum-averaged radiometric properties are averaged at wavelengths of 6.5–10 µm. The particle aggregate of consideration has the same structure as that shown in Figure 2a, but it has nine different position patterns of minor component particles, including the minor component particles gathering in the up, down, left, right, front, and back of the aggregate as well as the randomly scattered particles. In the figure, Ran.1 is the average result of three different random position patterns, Ran.2 is the average result of six different random position patterns, and Ran.3 is the average result of nine different random position patterns. These three results agree well with each other, indicating that, as the minor component particles are randomly distributed, their positions have a small influence on the radiometric properties of the aggregate.

In Figure 5, the minor component has a refractive index $m_2 = 3.102 + 0.0925i$ and is more absorptive than the major component ($m_1 = 1.6$), as it has a larger imaginary part of the refractive index than the major component. As seen in the figure, the radiometric parameters of the random position pattern are quite different from those of the other patterns. The absorption efficiency with the random position pattern is greater than those with the other patterns (up, down, left, right, front, and back), as shown in Figure 5b. Considering that in the random pattern the minor component particles are distributed more evenly in the aggregate compared to other patterns, the greater absorption efficiency with the random position pattern shows that evenly spreading the minor component particle in the aggregate can increase the radiation absorption of the aggregate. For each absorptive minor component particle, the absorbed energy is proportional to the incident energy [53]. With a random pattern, the minor component particles are evenly distributed in the aggregate, and the incident radiation reaching a minor component particle is less shielded by other minor component particles,

whereas, with other patterns in Figure 5, the minor component particles gather together, and before reaching a minor component particle the incident radiation may be well shielded by other minor component particles. Therefore, the average incident radiation energy for minor component particles with a random distribution is greater than that with other distributions in Figure 5, causing the absorbed energy by the aggregate with random patterns of minor component particles to be greater than that with other patterns.

The scattering efficiency with the random position pattern is lower than that with other patterns, as shown in Figure 5c. This may be because of the greater absorption by the minor component particles in the aggregate with the random position pattern than that with other position patterns; thus, more radiation is absorbed and less is scattered.

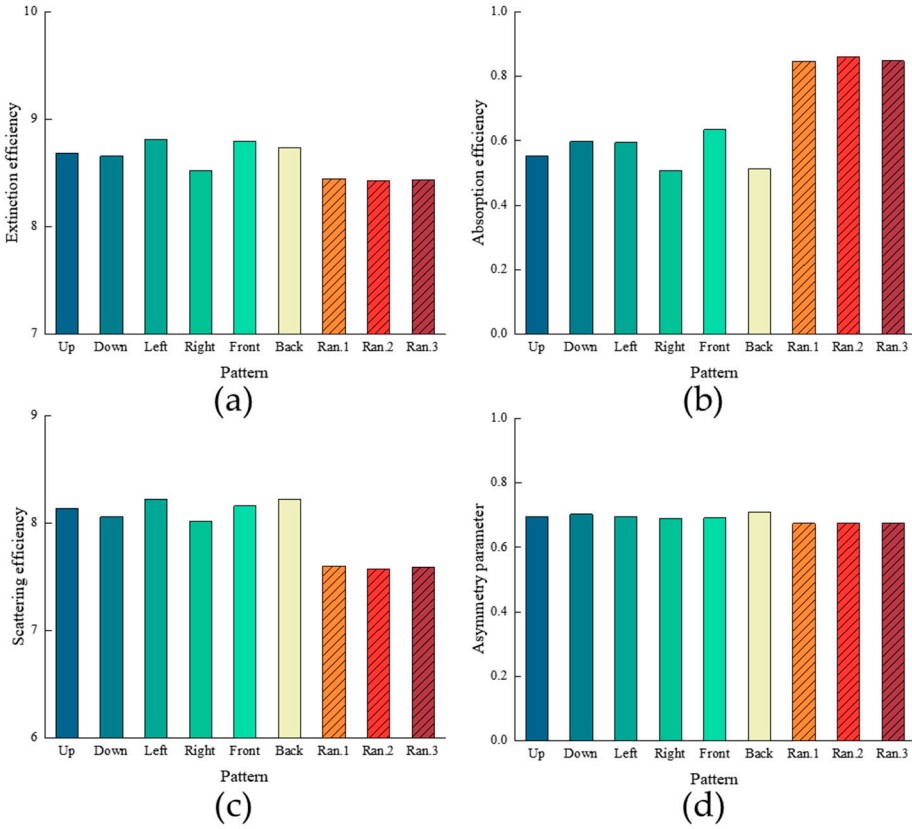

**Figure 5.** Radiometric properties for particle aggregates of Mode A with different position patterns: up, down, left, right, front, and back are for the minor component particles gathering in the up, down, left, right, front, and back in the aggregate as shown is Figure 2a–c; Ran.1, Ran.2, and Ran.3 are for the minor component particles randomly positioned. Ran.1 is the average result of three different random position patterns, Ran.2 is the average result of six different random position patterns, and Ran.3 is the average result of nine different random position patterns. The refractive indices of the components in the aggregates are $m_1 = 1.6$, $m_2 = 3.102 + 0.0925i$, and the particle radius is 2 μm. (**a**) Extinction efficiency; (**b**) absorption efficiency; (**c**) scattering efficiency; (**d**) asymmetric parameter.

For the extinction efficiency (Figure 5a), the random pattern is similar to the other patterns compared to the absorption efficiency (Figure 5b) and scattering efficiency (Figure 5c). This is because the extinction efficiency is the sum of the absorption efficiency and the scattering efficiency, and the two different deviations of the random patterns from the other patterns in Figure 5b,c offset each other; as a result, the extinction efficiency of the random pattern becomes more close to the other patterns in Figure 5a.

For the asymmetric parameter (Figure 5d), the random pattern has values that are similar to those of the other patterns, showing that the position pattern of the minor component particles has little effect on the ratio of forward scattering to total scattering.

### 3.3. Volume Fraction of Minor Components

Figure 6 shows that the radiometric properties changed with the volume fraction of the minor component for particle aggregates with refractive indices. The particle aggregate is of the structure in Figure 2a. The particle radius in these computations was set to 2 μm. The spectrum-averaged radiometric properties were averaged at wavelengths of 6.5–10 μm.

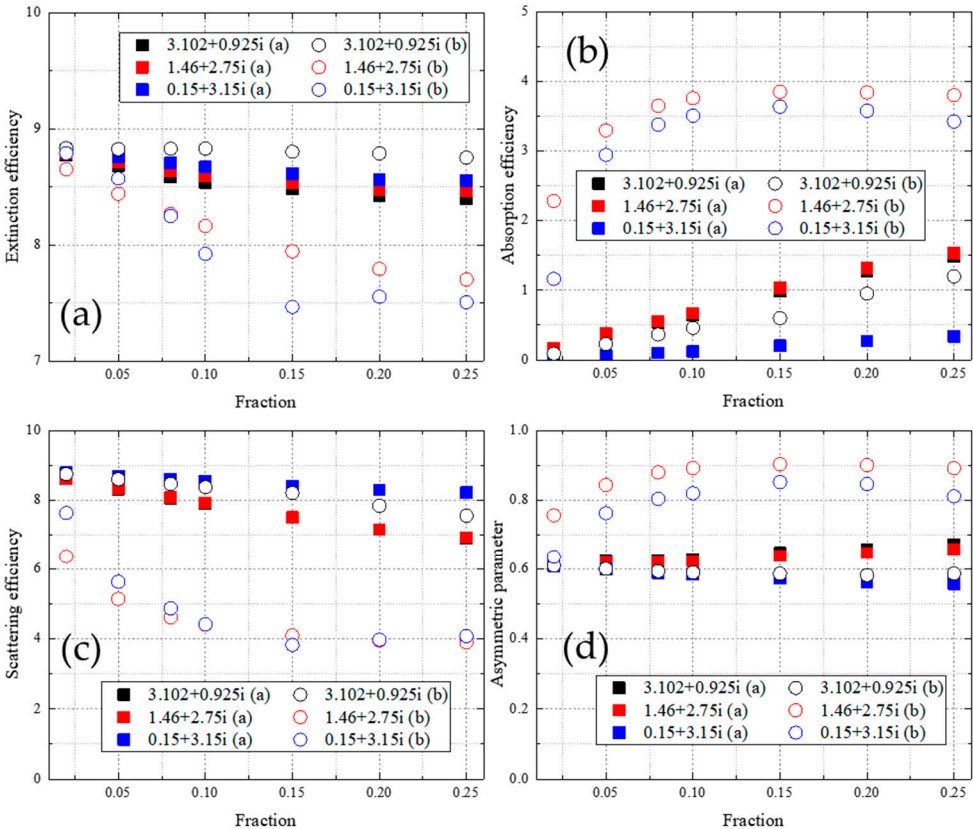

**Figure 6.** Radiometric properties of the particle aggregate changed with the volume fraction of the minor component of different refractive indices. The particle radius is 2 μm. (Square symbol: Mode A, with the minor component particles randomly positioned; circle symbol: Mode B; black symbol: minor component $m_2 = 3.102 + 0.0925i$; red symbol: minor component $m_2 = 1.46 + 2.75i$; blue symbol: minor component $m_2 = 0.15 + 3.15i$). (**a**) Extinction efficiency; (**b**) absorption efficiency; (**c**) scattering efficiency; (**d**) asymmetric parameter.

Figure 6b shows the absorption efficiency. The absorption efficiency generally increases with the volume fraction due to the increase in the volume of the minor component, which is more absorptive relative to the major component. In Mode B with $m_2 = 0.15 + 3.15i$ and $m_2 = 1.46 + 2.75i$, the absorption efficiency increases drastically as the volume fraction changes from 0.02 to 0.1 and levels off as the volume fraction exceeds 0.1, indicating that initially adding a small amount of the minor component into each particle of the aggregate can significantly increase the radiation absorption of the aggregate. However, in Mode A with $m_2 = 0.15 + 3.15i$ and $m_2 = 1.46 + 2.75i$, the absorption efficiency increases steadily with volume fraction. In Mode B, the minor component spreads evenly into each particle. As explained in Section 3.1, the absorption grows exponentially with optical length [1,52], while the EMA increases the effective optical length by the mean field approximation, which increases the weak

absorption of the major component, while in Mode A, the minor component only exists in a few particles, and radiation absorption is enhanced in a few spots in the aggregate. Increasing the volume fraction of the minor component or the number of minor component particles causes a steady increase in the absorption efficiency of the aggregate. For $m_2$ = 3.102 + 0.0925i, the minor component is slightly larger in absorption than the major component. Spreading the minor component into each particle (Mode B) has an effect on the increase in the absorption efficiency of the aggregate that is similar to that of concentrating it in a few particles (Mode A); therefore, both Mode A and Mode B show a steady increase in the absorption efficiency with the volume fraction.

Figure 6c shows the scattering efficiency. The scattering efficiency generally decreases with the volume fraction of the minor component, and the changes in scattering efficiency seem to be complementarily correlated to the changes in absorption in Figure 6b. The scattering efficiency drops significantly, while the absorption efficiency increases significantly as the volume fraction changes from 0.02 to 0.1. Figure 6d shows the asymmetric parameter. For the asymmetric parameter, the changes that occur with the increase in volume fraction are similar to those that occur with the increase in absorption efficiency in Figure 6b, showing that the asymmetric efficiency is closely related to absorption efficiency. Figure 6a shows the extinction efficiency, which is the sum of the absorption efficiency and scattering efficiency. The absorption efficiencies are similar in different aggregate modes with different refractive indices of the minor component; this is because the difference between Mode A and Mode B in the absorption efficiency in Figure 6a offsets that in the scattering efficiency in Figure 6b.

As shown in Figure 6a–d, Mode A has greater extinction efficiency, lower absorption efficiency, greater scattering efficiency, and lower asymmetric parameter values than Mode B when $m_2$ = 0.15 + 3.15i and $m_2$ = 1.46 + 2.75i, while both modes have similar values in all radiometric parameters when $m_2$ = 3.102 + 0.0925i. These phenomena are consistent with those obtained in Section 3.1; namely, the dispersion of a strong absorptive minor component in an aggregate can enhance absorption efficiency and decrease scattering efficiency.

## 4. Conclusions

The influence of component distribution, including the distribution mode and the minor component position pattern, on the radiometric properties of particle aggregates has been studied in this work. The main conclusions are as follows.

With the considered radii and wavelengths, the particle aggregates of Mode A have greater extinction efficiency, lower absorption efficiency, greater scattering efficiency, and smaller asymmetric parameter values than the aggregates of Mode B when $m_2$ = 0.15 + 3.15i and $m_2$ = 1.46 + 2.75i, while both modes have similar values in all radiometric parameters when $m_2$ = 3.102 + 0.0925i. This indicates that, when the minor component has an imaginary part of the refractive index larger than the major component, Mode A has greater extinction efficiency than Mode B, mainly due to the greater scattering of radiation caused by concentrating the minor component in a few particles. These particles contribute significantly to scattering of radiation. Since Mode B has a relatively even distribution of the minor component compared to Mode A, the difference in radiometric properties between these two modes suggests that evenly spreading the minor component, which is more absorptive than the major component, in the particle aggregate helps to increase the absorption efficiency, decrease the scattering efficiency, and ultimately decrease the extinction efficiency.

In Mode A, the positions of the minor component particles can also influence the radiometric properties of the particle aggregate. When the minor component particles are relatively evenly positioned in the aggregate or has an even position pattern (like a random position pattern), the particle aggregate has relatively high absorption efficiency, low scattering efficiency, and low extinction efficiency. This phenomenon is similar to that observed by the comparison of Modes A and B: as the minor component has a relatively large imaginary part of the refractive index and is more absorptive, evenly spreading the minor component in the particle aggregate helps to raise absorption and decrease extinction and the scattering of radiation in the aggregate.

**Author Contributions:** Conceptualization, Y.C.; Validation, Y.C.; Writing—original draft preparation, Y.C.; Writing—review and editing, Z.Y., Y.D.; Funding acquisition, Z.Y., Y.D.

**Funding:** This research was funded by National Basic Research Program of China (no. 2015CB251502) and the National Natural Science Foundation of China (nos. 51621062).

**Acknowledgments:** Dan Mackowski and Michael Mishchenko are gratefully acknowledged for making the MSTM and Mie code publicly available.

**Conflicts of Interest:** The authors declare no conflict of interest.

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
