# Peer review of "The Influence of Chemical Component Distribution on the Radiometric Properties of Particle Aggregates"

_applsci, doi:10.3390/app9071501_

Reviewer 1 Report

The manuscript entitled "Influence of chemical component distribution on radiometric properties of particle aggregates" by Chai, Yang, Duan deals with the problem of estimating the optical properties of aggregates formed through the traditional diffusion limited aggregation process. The approach followed by the authors is to evaluate the extinction and scattering efficiencies by means of the T-matric approach based upon a mean field approximation. 

They report numerical results for the scattering, extinction, absorption and asymmetry parameter as a function of the aggregate size and wavelength. The aggregates are composed by two different chemical species. 

My main concern about this work is the apparent lack of generality of the treatment and the corresponding results. The main issues are the following: 

- the authors arbitrarily choose very peculiar distributions of the different chemical species within the aggregate, without a clear physical expalanation of the general framework that makes it interesting for further insights. 

- the specific choice of the chemical species is not supported enough. Moreover, the discussion apparently brings to a trivial conclusion that the effect of the internal distribution is larger for the larger refractive index difference. 

- apparently, the results of the whole manuscript are obtained on the basis of a very limited statistics. The authors should clearly explain why the results can be considered reliable basing upon three position patterns only

- Finally, the overall manuscript appears as a acritic report of a few numerical simulations, without any physical discussion about the dependence of the results on the distribution of the particles inside the aggregates, in contrast with the results obtained adopting the mean field approximation. This conlcusion is not clearly assessed at all, and in the present form of the manuscript cannot be safely assesed for the reasons described above.

These criticisms suggest to reject the manuscript in its present form. In the case the authors can appreciably improve the previous points, a new manuscript should be prepared accordingly, discussing the validity of the assessment of above. 

Author Response

The authors thank the reviewer for the suggestion.

The details are in the Word file

Reviewer 2 Report

General comments

The base reasoning of the research must be presented, otherwise is only about playing with numbers. What are the problems to be solved? Is there any practical application?

The discussion is absent. In fact, the justification for the results must be done by mention others researches. There is no a single reference during the discussion of results.

There are many sentences without the necessary references. In the next section some examples are presented.

Specific comments

Line 28 - “The radiometric properties of particle aggregates mainly include the extinction efficiency, absorption efficiency, scattering efficiency, and asymmetric parameter.”

Please, the appropriate references.

Line 39 - “The effective-medium approximations (EMA) has been widely used to model the radiometric properties of homogeneous substances.”

Please, the appropriate references.

Line 55 - “For spherical particle aggregates with complex structures, the Multi-Sphere T-Matrix (MSTM) method is used to calculate the radiometric properties.”

Please, the appropriate references.

Line 93 - “The particle radius R is set to 2-5 μm; this particle radius range is common for a number of real particles such as the comet dust or the smoke particles.”

Please, the appropriate references.

Line 101 - “In mode A, a particle is composed either of the major component or the minor component; in this model, the minor component is concentrated in a few particles, like in many natural agglomerations or particles [23].”

Please, explain the difference between major and minor component. There is not scientifically correct the expression “the minor component is concentrated in a few particles”.

Line 104 - “Figure 1(a) and (d) show a particle aggregate of mode A and mode B, respectively; the two modes have different distributions of the minor component”

Please, check this sentence. Mode A don’t have minor component; all the particles have equal composition.

Line 133 - “The optical refractive index of the major component particles m1=1.6, the value for typical calcite or dolomite.”

Please, the appropriate references.

Line 144 - “The effective refractive index meff of the spherical particles is:”

Please, the appropriate references.

Figures

Figure 4 - Check the mention to this figure (Fig 3 along the text).

Figure 5 - The size of the lettering is too small.

Author Response

The authors thank the reviewer for the suggestion. 

The details are in the Word file.

Reviewer 3 Report

More detailed description of the model used to generate the results shown in Figures 1 and 2 has to be presented. There are a lot of results generated, but very little description of the numerical details of the modeling approach.

Formatting inconsistencies are present, please proofread the paper carefully.

Author Response

The authors thank the reviewer for the suggestion.

The details are in the Word file.

Round  2

Reviewer 1 Report

I still maintain my former idea that the manuscript results are quite limited in terms of statistical strength. Nevertheless I agree that improvements have been done, and the manuscript can be published after minor revisions listed here below. 

- Figure 4 and 6. Please represent data as separate points corresponding to each result, with error bars if possible, instead of continuous lines. 

- Figure 5 must be ameliorated and set properly for publication  Please better clarify the meaning of the plots briefly in the caption. 

Author Response

The authors thank the reviewer for the suggestion. The details are in the word file.

Reviewer 2 Report

Authors improved the manuscript according my comments.

Despite discussion could be deeper, the manuscript can go to the next stage.

Author Response

The authors thank the reviewer for the suggestion. The discussion of physical mechanism would be deeper in the future work.